# The Effects of Arginine-Based Supplements on Fatigue Levels following COVID-19 Infection: A Prospective Study in Romania

**DOI:** 10.3390/healthcare11101477

**Published:** 2023-05-18

**Authors:** Adina Turcu-Stiolica, Claudiu Marinel Ionele, Bogdan Silviu Ungureanu, Mihaela-Simona Subtirelu

**Affiliations:** 1Pharmacoeconomics Department, University of Medicine and Pharmacy of Craiova, 2-4 Petru Rares Street, 200349 Craiova, Romania; 2Doctoral School, University of Medicine and Pharmacy of Craiova, 2-4 Petru Rares Street, 200349 Craiova, Romania; 3Gastroenterology Department, University of Medicine and Pharmacy of Craiova, 2-4 Petru Rares Street, 200349 Craiova, Romania

**Keywords:** fatigue, COVID-19, arginine, persistent symptoms, post-acute COVID-19 syndrome

## Abstract

The purpose of this study was to examine the effects of two arginine-based supplements on the fatigue level of patients following the COVID-19 infection. This is a prospective study of the SARS-CoV-2-infected patients divided into two groups (according to family physicians’ prescriptions, Group 1 of patients were treated with Astenor Energy^®^ containing arginine aspartate, B6 vitamin, biotin and magnesium, and Group 2 of patients were treated with Astenor Forte^®^ containing L-arginine and malic acid). The patients visited their family physicians from October 2021 to January 2022, complaining of physical and/or mental fatigue following the COVID-19 infection. We recorded 505 patients (146 patients in Group 1 and 359 patients in Group 2) and analyzed the fatigue level using the Fatigue Assessment Scale (FAS) through its total (FAS-T), mental (FAS-M) and physical (FAS-P) scores, at baseline and after three months of treatment. There was no difference between the two groups in terms of age (*p* = 0.265), but more men were included in Group 1 than in Group 2 (*p* = 0.001). The patients from Group 2 were significantly more likely to be treated at home than those included in Group 1 (89.7% vs. 65.1%, *p* < 0.0001) because of the lower severity of the COVID-19 infection (asymptomatic or mild: 82.5% vs. 48.7%, *p* < 0.0001). After 3 months of treatment, patients indicated no fatigue in the higher percentage compared to than at the baseline (68.7% vs. 27.7%), and the fatigue level significantly decreased both in Group 1 (median baseline 33.0 vs. follow-up 17.00, *p* < 0.0001) and Group 2 (median baseline 25.0 vs. follow-up 17.00, *p* < 0.0001). These findings suggest that supplements with L-arginine may be proposed as a remedy to restore physical and mental performance affected by the fatigue burden in people with COVID-19 or following the COVID-19 infection.

## 1. Introduction

A large proportion of patients who have been diagnosed with the COVID-19 virus report the long-term clinical sequelae (weeks or months) after the onset of symptoms [1,2]. These post-COVID symptoms may include gastrointestinal and neurological signs and symptoms, such as dyspnea, fatigue, arrhythmias, heartburn and difficulty with memory and attention (“brain fog”), with a substantial impact on the quality of life [3,4]. Fatigue and attention difficulties may also be associated with reduced physical function, which may prevent full resumption of daily activities before the infection [5,6]. Several processes are currently being investigated for their involvement in the pathophysiology of prolonged COVID, including viral persistence, chronic inflammation, autoimmunity, disruption of metabolic and redox homeostasis and endothelial dysfunction [7,8].

The heterogeneity and multitude of clinical manifestations of long-term COVID has so far prevented the development of specific treatments for the condition, so that its management is largely based on symptomatic treatments and healthy lifestyle recommendations. Decreased immunity of patients is a significant risk factor for infection with respiratory viruses. Adequate nutrition and good nutritional status are viewed as important elements for an optimal immune response to prevent infections [9]. Many patients in several studies [10] were deficient in selenium, but also in more than one nutrient such as vitamin B6, magnesium and malic acid. Therefore, the nutritional deficiencies may possibly favor the onset of COVID-19 and increase the severity of the disease [11].

L-arginine is a key regulator of the immune, respiratory and endothelial systems. Its properties are regulated by two main metabolizing enzymes: nitric oxide (NO) synthase and arginase [10]. The flow of L-arginine toward NO synthesis is associated with the beneficial effects it has on the immune and vascular health, while its catabolism to ornithine by arginase has been associated with an abnormal immune response and endothelial dysfunction [12]. Studies so far indicate that L-arginine metabolism is altered in patients with COVID-19 [13,14]. During the COVID-19 infection, upregulation of arginase activity reduces the circulating levels of L-arginine and shifts its metabolism away from NO production to induce immune and endothelial dysfunction, inflammation and thrombosis, which ultimately leads to vascular occlusion and multiorgan failure [15]. L-arginine supplementation restores the proliferative capacity of T cells obtained from patients with the acute respiratory distress syndrome during COVID-19 [16]. Furthermore, oral L-arginine supplementation has been shown to reduce the need for respiratory support and length of hospital stay in patients with severe COVID-19 [17]. Finally, oral supplementation with L-arginine reduced the burden of persistent symptoms and improved perceived exertion in a large cohort of patients with long-term COVID [18]. Moreover, L-arginine has shown no adverse side effects in patients when taken in moderation [19]. L-arginine and L-aspartate seem to have induced synergistic metabolic effects. L-arginine might have reduced lactic acid production by the inhibition of glycolysis, and L-aspartate may have favored fatty acid oxidation. In addition, the results indicate improved work efficiency after L-arginine and L-aspartate intake [20].

Nutritional supplementation with L-arginine and L-aspartate leads to an increase in fat oxidation and reduces blood lactate and associated oxygen consumption and heart rate and ventilation during submaximal cycle exercise. This may involve exercise tolerance, which may have important implications for patients diagnosed with COVID-19 [21]. A low L-arginine–ornithine ratio has been observed in patients with COVID-19 [22], and this indicates an increase in arginase activity in these patients. In another study, plasma L-arginine levels were shown to correlate inversely with the severity of COVID-19 [23]. This study also showed that the expression of the activated GP IIb/IIIa complex, known to be involved in platelet activation [23,24], is higher in patients with severe COVID-19 compared to healthy patients and inversely correlated with the plasma concentration of L-arginine [25].

In fact, a decrease in L-arginine bioavailability has been shown to cause diminished T-cell response and function, ultimately leading to increased susceptibility to infection [26,27]. Twelve weeks of L-arginine supplementation significantly decreased IL-21 levels [28], whereas NO has been shown to suppress the proliferation and function of human Th17 cells [29].

Previous studies have shown that L-arginine supplementation improves respiratory function and exercise tolerance in patients with lung disease [30] and those with congestive heart failure [31], as well as heart transplant recipients [32]. Additionally, L-arginine supplementation can increase aerobic and anaerobic performance in healthy adults. However, other studies have found no effects of L-arginine supplementation on human performance [33,34].

Fatigue is described as a nonspecific symptom that is prevalent in many categories of patients, and it is an important component of many physical diseases [35]. For example, fatigue is a common symptom experienced by patients having chronic diseases such as cancer [36]. Hence, several fatigue questionnaires have been used for specific populations such as cancer patients [37,38]. Fatigue also plays an important role in the healthy population. Severe fatigue during a relatively long period can lead to sick leave and work disability.

The aim of this study is to examine the fatigue level after treatment with L-arginine in patients having fatigue symptoms following COVID-19 infection.

## 2. Materials and Methods

### 2.1. Study Design

We conducted a 12-week observational prospective study using patients who visited family physicians from October 2021 to January 2022, adopting the protocol drawn up by the Declaration of Helsinki and approved by the University of Medicine and Pharmacy of Craiova Ethics Committee no. 175/29 October 2021. All subjects signed the informed consent for inclusion before they participated in this study and responded to the questionnaire.

This study included patients aged at least 18 years old who had been infected with the SARS-CoV-2 virus complaining of mental (lack of mental clarity, decreased attention, difficulty concentrating, memory loss) or/and physical (lack of energy for daily activity, decreased muscle strength) fatigue symptoms during or after COVID-19 infection. We excluded patients with any psychological symptoms prior to COVID-19 infection.

As family physicians decided, patients with high levels of alanine aminotransferase (ALT) and aspartate aminotransferase (AST) were included in Group 1 and received for three months, 10 days/month, the treatment with Astenor Energy^®^ (950 mg arginine aspartate, 4 mg vitamin B6, 150 mcg biotin and 83.3 mg magnesium). Patients with normal levels of ALT and AST were included in Group 2 and received Astenor Forte^®^ (413 mg L-arginine and 1500 mg malic acid) for three months, 10 days/month. Following the three months period of treatment, the patients from Group 1 measured their ALT and AST to be compared with the baseline values.

Patients’ characteristics (age and sex, Body Mass Index (BMI), ALT, AST, comorbidities), COVID-19 characteristics (severity, treatment) were collected. The data collected included adverse events (AEs). All patients completed the Fatigue Assessment Scale (FAS) baseline, before starting the treatment, and after the three months of treatment.

The Fatigue Assessment Scale (FAS) is a unidimensional fatigue measurement method which comprises 10 questions that describe how a person generally feels. With regard to convergent validity, it was anticipated that the FAS would have high associations with related fatigue measures and permit us to examine the dimensionality of fatigue [39]. Previous studies provided the information regarding the FAS used in patients diagnosed with sarcoidosis [40].

FAS has been used for measuring the fatigue in many diseases, which confirms its reliability and validity [41,42]. The ten items of the questionnaire reflect both mental and physical fatigue, with a 5-point scale response, so the total FAS score ranges from 10 to 50. A respondent has no fatigue (normal) if FAS score is equal to or less than 21; otherwise, the respondent has substantial fatigue (score 22–34) or extreme fatigue (score ≥ 35) [34]. The Romanian version of FAS was used in this study [40,43].

### 2.2. Statistical Analysis

All statistical analyzes were performed using GraphPad Prism (v 9.5.1), (GraphPad Software, LLC, Boston, MA, USA), and statistical significance was considered two-tailed at *p* < 0.05. Means with standard deviations (SD) and medians with interquartile ranges (IQR) were reported as appropriate for continuous variables. The discrete variables were expressed as number and percentage. In the case of continuous characteristics, after verifying their distributions, the *t*-test or Mann–Whitney test was used to analyze the differences between Group1 and Group 2 outcomes or between baseline and the following period. In the case of discrete variables, the chi-square test was applied to compare the outcomes between the two groups. Kruskal–Wallis test was used to compare continuous variables according to more than 2 groups of patients (grouping by severity). Pearson’s or Spearman’s correlation coefficients were calculated to determine correlations between variables and visually presented with a correlation heatmap (colors range from bright blue for strong positive correlations to bright red for strong negative correlations). The significance level lower than 0.05, using two-sided test, was adopted.

## 3. Results

As reported in Table 1, the patient population was composed of 146 patients included in Group 1 and 359 patients included in Group 2. There was no difference between the two groups in terms of age, but more men were included in Group 1 than in Group 2. The patients from Group 2 were significantly more likely to be treated home than those included in Group 1 (89.7% vs. 65.1%, *p* < 0.0001) because of lower severity of the COVID-19 infection (asymptomatic or mild: 82.5% vs. 48.7%, *p* < 0.0001).

The mean for FAS score was 27.69 ± 9.34 (range 10–49) for all 505 participants. No differences were found between men and women (total score mean ± SD: 27.3 ± 9.39 vs. 28.01 ± 9.31, *p*-value = 0.462; physical score mean ± SD: 14.6 ± 4.7 vs. 14.9 ± 4.36, *p*-value = 0.496; mental score mean ± SD: 12.7 ± 5.1 vs. 13.09 ± 5.4, *p*-value = 0.49). Low but significant positive correlation was found between age and the level of fatigue (Spearman’s rho = 0.203, *p*-value < 0.0001) and between BMI and fatigue (Spearman’s rho = 0.232, *p*-value < 0.0001).

The median physical fatigue score was significantly higher than that for mental fatigue (14.77 ± 4.51 vs. 12.91 ± 5.26, *p* < 0.0001). Kruskal–Wallis test was used to demonstrate the statistically significant difference in the total score, physical and mental scores across the four types of COVID-19 infection’s severity, as listed in Table 2, with the lowest values for asymptomatic patients and the highest values for the patients with severe COVID-19 infection.

Of 505 participants, 105 (21%) reported fatigue following the COVID-19 infection and 204 (40%) reported fatigue both during and following the COVID-19 infection. Evaluating correlations with the severity of initial infection and fatigue following the COVID-19 infection was dependent of infection severity (Spearman’s rho = 0.338, *p*-value < 0.0001). The same positive correlation was found among the 109 patients that reported fatigue during the COVID-19 infection (Spearman’s rho = 0.387, *p*-value < 0.0001), as shown in Figure 1. Significant positive correlations were also found between fatigue and the number of comorbidities (Spearman’s rho = 0.384, *p*-value < 0.0001).

No differences in the fatigue onset between the two groups were observed. As shown in Table 3, there were patients with fatigue during the COVID-19 infection (38% in Group 1 and 39% in Group 2), post COVID-19 infection (25% in Group 1 and 19% in Group 2) and in both periods, during and post COVID-19 infection (38% in Group 1 and 41% in Group 2). Chi-squared test indicated that there were significant (*p* < 0.0001) differences between the fatigue types: patients in Group 1 experienced both more physical and mental fatigue than patients in Group 2, whereas patients in Group 2 experienced only more physical fatigue than the patients in Group 1.

A finding of a fatigue score measured by FAS below 21 means that the fatigue levels of the patients were not significant, and that some coping strategies have been effective. According to FAS levels, at baseline, patients had substantial fatigue (43.4% significant fatigue, 28.9% extreme fatigue). After 3 months of treatment, patients indicated no fatigue in the higher percentage compared to that at the baseline (68.7% vs. 27.7%).

Age was significantly correlated with both physical (Spearman’s rho = 0.213, *p* < 0.0001) and mental (Spearman’s rho = 0.196, *p* < 0.0001) fatigue levels. Fatigue significantly correlated with BMI (Spearman’s rho = 0.232, *p* < 0.0001), severity (Spearman’s rho = 0.387, *p* < 0.0001), the number of comorbidities (Spearman’s rho = 0.384, *p* < 0.0001), ALT (Spearman’s rho = 0.245, *p* < 0.0001) and AST (Spearman’s rho = 0.317, *p* < 0.0001).

After evaluating the non-Gaussian distribution of FAS through the violin graphs, the Mann–Whitney test indicated that fatigue levels were significantly (*p* < 0.0001) reduced, as shown in Figure 2, in both groups: FAS median in Group 1 decreased from 33.0 to 17.0 and in Group 2 from 25.0 to 17.0.

The same trend was observed for AST and ALT not only after the treatment in Group 1 but also in Group 2, as shown in Figure 3.

The safety profile of Astenor ENERGY^®^ and Astenor FORTE^®^ was as expected, no AEs were observed.

## 4. Discussion

In the current clinical study, we showed that L-arginine supplementation reduced fatigue in adults during and following COVID-19. These findings support the view that increasing NO bioavailability through the synergistic effects of L-arginine ameliorates the post-acute sequelae of COVID-19.

Qualitative studies to date have examined wider patient experience. They revealed several common themes, including symptom variability, physical and intellectual fatigue, patients’ burden of accessing care and fear of a permanent reduction in physical and cognitive abilities [16,17,18,19,20,21,22,23,24,25,26,27]. Cognitive complaints and persistent fatigue have been reported in the literature among the most debilitating symptoms for patients diagnosed with COVID-19. However, our understanding of how patients experience and make sense of these symptoms in their daily lives is still in its infancy.

Townsend et al. [44] highlighted the importance of assessing the post-viral fatigue in patients recovering from the COVID-19 acute phase, independent of the severity of infection. On the contrary, our study did find that post-viral fatigue was correlated with the severity of the COVID-19 infection.

Similar to our results, Townsend [44], Kamal [45] and Nehme [46] described that older patients have a higher risk of suffering from fatigue, but contrary to them, we found no differences between male and female patients.

In our study, we found that patients in Group 2 were more likely to be treated at home that patients in Group 1 (89.7% vs. 65.1%) most of all because of the lower severity of the COVID-19 infection. Thus, it seems that SARS-CoV-2 can cause viral hepatitis, but it can also cause an increase in ALT and AST, which can increase the severity of the disease [16].

Professionals need to understand how patients themselves describe their symptoms and their impact on their lives in order to help the patients to manage their condition. Beyond disease management and validation of patients’ worries, a good insight into their ailment is needed.

The available data support the notion that the hepatic injury in the SARS-CoV-2 infection is a consequence of a multifactorial attack [40]. It has been observed in many studies from the beginning of the pandemic that the results of the liver function tests (LFTs) can be elevated [47]. Biological analyses have shown that the incidence of liver injury ranged from 14% to 53%, mainly indicated by the hepatocytes elevation of AST and ALT [45,46,47,48]. In our study, we assessed AST and ALT, and the results that we have obtained are in line with those in the literature [49]. However, during follow-ups, patients with higher transaminases and Astenor Energy^®^ regimen had decreased AST and ALT levels after a 3-month lapse.

In our study, the fatigue after 3 months of treatment dropped in both groups (from 39% to 19%, especially in Group 2). Moreover, patients in Group 1 experienced more physical and mental fatigue than patients in Group 2, whereas patients in Group 2 experienced more physical fatigue alone than patients in Group 1. This can be associated with the fact that Group 1 had more severe cases of COVID-19, some of them even treated in the hospital for longer periods of time. According to Ceban [5], approximately 63% of individuals with sequelae of COVID-19 have difficulty performing daily tasks and self-care and also have a social impairment with difficulty returning to work.

Arginine was demonstrated to be an attractive alternative for patients with mild to moderate forms of fatigue for various reasons. First, arginine is more psychologically accepted because it is perceived as a nutrient rather than a drug [50]. Moreover, arginine was found to exert synergistic effects when administered concomitantly with other supplements such as malic acid, magnesium, biotin and vitamin B6 [31].

Andrade Silva et al. [51] and Wu et al. [52] demonstrated a significant reduction in malic acid in severe and mild COVID-19 patients. Astenor Forte^®^ also contains malic acid involved in the Krebs cycle and energy metabolism, helping the recovery after COVID-19.

The current study may have some limitations that should be considered when interpreting the results. Due to the relatively small number of participants and the single-center nature of this study, our results should be considered to be preliminary. Further studies with larger populations, conducted in more centers and using different study methodologies (longer intervention with supplements, cross-over design) are warranted to confirm these promising findings. However, this study did not aim to determine specific nutritional deficiency, but rather to show the general trend of deficiency and the importance of supplementation in the cases of the COVID-19 infection. Interviews before and after 3 months of treatment were also carried out by the family doctors, in which they assessed fatigue, BMI, ALT and AST levels, comparing them with the initial values. One of the limitations observed is the fact that the family doctors did not quantify the results from the interviews regarding the improvement of nutrition and reintegration into the family life and work, although it was observed that the patients after treatment reported low levels of both fatigue and transaminases.

Our findings regarding the total of 40% of the patients that reported fatigue as a general symptom both during the COVID-19 infection and following it corresponded to those reported by Stavem et al. The authors of [53] found that 46% of the respondents reported fatigue about 4 months after the symptom onset; these represent a higher prevalence than in the normal population. Moreover, the World Health Organization described “pandemic fatigue” as a feeling of distress in the population that reacts to the prolonged period of uncertainty during the pandemic [52].

We can say that the impact of a new, unknown and lethal virus cannot be underestimated. It is imperative to understand the importance of the social effects of COVID-19 and also to know how they can affect both the physical and mental health [54].

The high prevalence of fatigue following the cases of COVID-19 is likely to have important consequences for patients, although the observation period was only 3 or 6 months. It is possible that specific interventions or rehabilitation could influence the development of these symptoms; however, such interventions should be evaluated in randomized controlled trials [55]. Those who are the most at risk of experiencing persistent fatigue or other symptoms may benefit from supplements and a personalized regimen [53,56].

Another limitation of our study was in simply assessing the presence of persistent symptoms; we could only find out if those symptoms were reduced after taking the recommended supplements.

Accordingly, we can say that L-arginine supplements are a safe practice to restore physical and mental performance affected by the fatigue burden in people with COVID-19 and after it. As a supplement, it could be recommended by community pharmacists or through remote consultations during the COVID-19 pandemic [48,49,50,51,52,53,54,55,56,57,58].

## 5. Conclusions

L-arginine supplementation improved fatigue in adults with COVID-19. The combination of L-arginine and aspartate, biotin, acid malic and vitamin B6 may therefore be proposed as a remedy to restore physical performance and relieve persistent symptoms in people with COVID-19 and following the COVID-19 infection.

## Figures and Tables

**Figure 1 healthcare-11-01477-f001:**
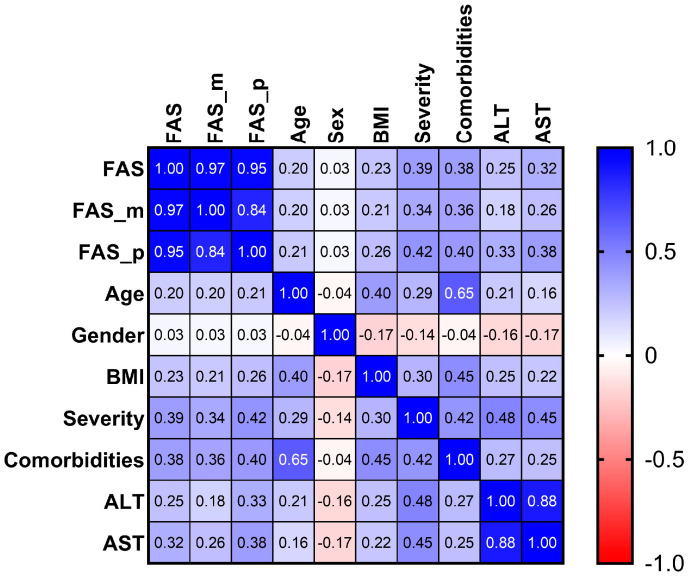
The correlation heatmap between FAS scores and measured characteristics (colors range from bright blue for strong positive correlations, to bright red, for strong negative correlations).

**Figure 2 healthcare-11-01477-f002:**
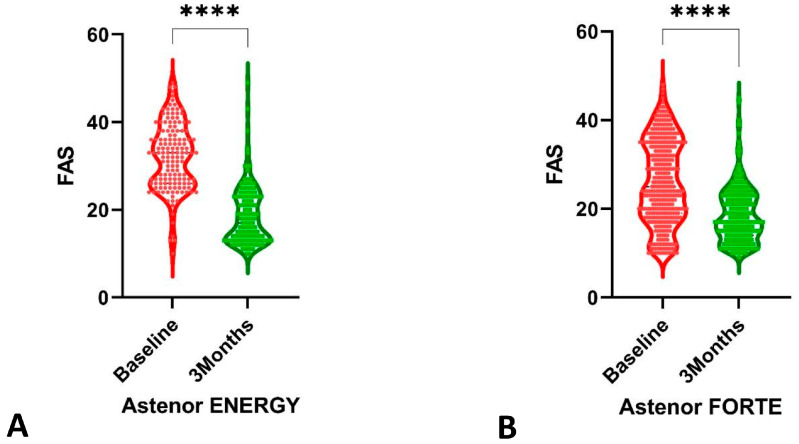
The comparison between baseline and after 3 months of treatment for fatigue levels: (**A**) treatment with Astenor ENERGY^®^; (**B**) treatment with Astenor FORTE^®^. **** *p*-value < 0.0001.

**Figure 3 healthcare-11-01477-f003:**
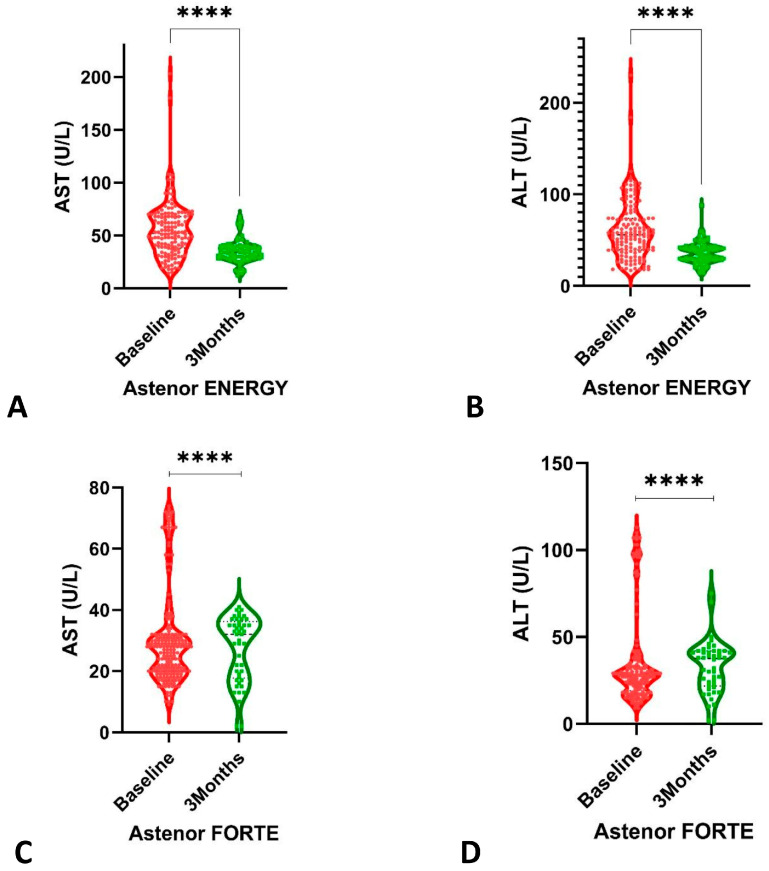
The comparison between baseline and after 3 months of treatment for AST and ALT: (**A**,**B**) treatment with Astenor Energy; (**C**,**D**) treatment with Astenor Forte. **** *p*-value < 0.0001.

**Table 1 healthcare-11-01477-t001:** Demographics and baseline characteristics.

Patients’ Characteristics	Total Cohort (*n* = 505)	Group 1 (*n* = 146)	Group 2 (*n* = 359)	*p*-Value
Age, years, median (IQR)	50 (39–63)	53 (29.75–66)	49 (39–61)	0.265
Sex, male, *n* (%)	231 (45.7%)	84 (57.5%)	147 (40.9%)	0.001
Severity infection				<0.0001
Asymptomatic, *n* (%)	14 (2.8%)	3 (2.1%)	11 (3.1%)	
Mild, *n* (%)	353 (69.9%)	68 (46.6%)	285 (79.4%)	
Moderate, *n* (%)	120 (23.8%)	59 (40.4%)	61 (17%)	
Severe, *n* (%)	18 (3.6%)	16 (11%)	2 (0.6%)	
Treatment				<0.0001
In specialty, *n* (%)	40 (7.9%)	31 (21.2%)	9 (2.5%)	
ICU, *n* (%)	4 (4%)	4 (2.7%)	0	
Ambulator, *n* (%)	44 (8.7%)	16 (11%)	28 (7.8%)	
Home, *n* (%)	417 (82.6%)	95 (65.1%)	322 (89.7%)	
BMI baseline (kg/m^2^), median (IQR)	27 (24–30)	29.1 (24.1–31.4)	26 (24–30)	<0.0001
BMI follow-up (kg/m^2^), median (IQR)	27 (24–30)	29 (24.1–31.4)	26 (23.9–30)	<0.0001
ALT baseline, median (IQR)	33 (24–61.25)	56 (39.3–74)	28 (19–33.3)	<0.0001
ALT follow-up, median (IQR)	35.50 (27–42)	52 (38–68)	28 (20–32)	<0.0001
AST baseline, median (IQR)	32 (23–58)	34.5 (29–42)	37.5 (21.8–42)	0.287
AST follow-up, median (IQR)	33 (28–38)	34 (29–39)	32 (17.8–36.3)	0.005
Comorbidities, median (IQR)	1 (0–1)	1 (0–1)	1 (0–1)	0.785
Comorbidities				<0.0001
No comorbidities, *n* (%)	235 (46.5%)	44 (22.4%)	191 (43%)	
Cardiovascular, *n* (%)	74 (14.7%)	30 (15.3%)	44 (9.9%)	
Diabetes, *n* (%)	67 (13.3%)	29 (14.8%)	38 (8.6%)	
Hypertension, *n* (%)	132 (26.1%)	34 (17.3%)	98 (22.1%)	
Respiratory, *n* (%)	34 (6.7%)	15 (7.7%)	19 (4.3%)	
Cancer, *n* (%)	6 (1.2%)	3 (1.5%)	3 (0.7%)	
Hepatic diseases, *n* (%)	40 (7.9%)	25 (12.8%)	15 (3.4%)	
Others, *n* (%)	52 (10.3%)	16 (8.2%)	36 (8.1%)	

IQR, interquartile range; ICU, intensive care unit; BMI, Body Mass Index; ALT, alanineaminotransferase; AST, aspartateaminotransferase.

**Table 2 healthcare-11-01477-t002:** Variation of the patients’ fatigue according to severity of COVID-19 infection.

Patients Fatigue According to Severity of Infection	Median (Interquartile Range)	*p*-Value
Total score		<0.0001
Asymptomatic	20.50 (18–27.25)	
Mild	24 (19–34)	
Moderate Severe	33 (28–38) 40.50 (35.50–43.25)	
Physical score		<0.0001
Asymptomatic Mild	9.50 (7.75–12.50) 11 (8–16)	
Moderate Severe	16 (12–19) 20 (17–21)	
Mental score Asymptomatic Mild Moderate Severe	11.50 (9–16) 14 (11–17) 18 (16–19) 20.50 (18.75–23)	<0.0001

**Table 3 healthcare-11-01477-t003:** Characteristics of the patients’ fatigue.

Patients Fatigue	Total *n* = 505	Group 1 *n* = 146	Group 2 *n* = 359	*p*-Value
Fatigue type				<0.0001
Physical fatigue	174 (34.4%)	24 (16%)	150 (42%)	
Mental fatigue	4 (0.8%)	1 (1%)	3 (1%)	
Physical and mental fatigue	327 (64.8%)	121 (83%)	206 (57%)	
Fatigue onset				0.3216
During COVID-19 infection Post COVID-19	196 (38.8%) 105 (20.8%)	55 (38%) 37 (25%)	141 (39%) 68 (19%)	
During and post COVID-19	204 (40.4%)	56 (38%)	148 (41%)	
Fatigue levels at baseline No fatigue (score 10–21) Fatigue (score 22–34) Extreme fatigue (score ≥ 35)	140 (27.7%) 219 (43.4%) 146 (28.9%)	7 (4.8%) 83 (56.8%) 56 (38.4%)	133 (37%) 136 (37.9%) 90 (25.1%)	<0.0001
Fatigue levels at follow-up No fatigue (score 10–21) Fatigue (score 22–34) Extreme fatigue (score ≥ 35)	347 (68.7%) 148 (29.3%) 10 (2%)	99 (67.8%) 44 (30.1%) 3 (2.1%)	248 (69.1%) 104 (29%) 7 (1.9%)	0.9615

## Data Availability

The data can be requested from the corresponding author for proper reasons.

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
