# Peer review of "The Effects of Arginine-Based Supplements on Fatigue Levels following COVID-19 Infection: A Prospective Study in Romania"

_healthcare, 2023, doi:10.3390/healthcare11101477_

Round 1
Reviewer 1 Report
This study is designed to examine the effects of two arginine-based supplements on the fatigue level of patients following COVID-19 infection. They found that after 3 months of treatment, in both groups, the overall fatigue level decreased significantly compared to the base level. Their findings suggest that supplements with L-arginine could be a potential remedy to restore physical and mental performance affected by fatigue burden in people with COVID-19 or following COVID-19. Though this study has merit, the authors seem rushed to submit the manuscript without substantial editing.
Specific comments:
- The authors performed this study that answered the research question.
- Overall, this paper needs substantial editing of grammar and language before being published. A few examples,
o In line 66, “A low L-Arginine-ornithine ratio has been observed in patients with COVID-19 [18] and this indicates an increase in arginase activity in these patients.” A comma (,) would be best before the conjunction “and.”
3. Any explanation on the exclusion of the symptom improvement other than L-arginine would be better. How do you negate that the fatigue was improved as a result of the food items they were taking?
4. Importantly, the overall time of this study is very short. Any explanation would convince the readers about the short time period and relevant possible reasons.
5. The introduction is not sufficient enough to gain the overall background of this study. Adding a bit more relevant literature studies in the introduction will enhance the understanding of the readers.
Author Response
We are very grateful to the constructive comments from you. We also thank you for the time and effort on reviewing our manuscript. We have carefully addressed point-by-point all the comments and made corrections in our manuscript using tracked changes.
Specific comments:
- The authors performed this study that answered the research question.
- Overall, this paper needs substantial editing of grammar and language before being published. A few examples,
- In line 66, “A low L-Arginine-ornithine ratio has been observed in patients with COVID-19 [18] and this indicates an increase in arginase activity in these patients.” A comma (,) would be best before the conjunction “and.”
Thank you for the suggestion, all the manuscript was checked and rephrased where needed by a native English speaker. Line 66 – modified.
- Any explanation on the exclusion of the symptom improvement other than L-arginine would be better. How do you negate that the fatigue was improved as a result of the food items they were taking?
Thank you for this valuable question and insight.
First of all, the patients included in the study were interviewed by a general practician after being diagnosed with COVID-19 infection and accusing mental or/and physical fatigue symptoms. The general practician decided how to manage these symptoms and food items they could take were not discussed with the patients. We agree proper nutrition could improve fatigue and we verified if significant differences in BMI values between baseline and follow-up were registered. No significant differences were observed, as in Table 1.
It is well known that immunity and nutrition have played an important role in the coronavirus infection. A proper diet and good nutritional status are considered important elements for optimal immune response to prevent infections [Calder et al. 2020].
Evidence suggests that nutrients are involved in the development of COVID-19 [Im et al. 2020], but our study does not aim to measure the improvement after food intake, the nutritional status of COVID-19 patients was unknown both baseline and after 3 months.
References
Calder P.C., Carr A.C., Gombart A.F., Eggersdorfer M. Optimal Nutritional Status for a Well-Functioning Immune System Is an Important Factor to Protect against Viral Infections. Nutrients. 2020;12(4).
Im JH, Je YS, Baek J, Chung MH, Kwon HY, Lee JS. Nutritional status of patients with COVID-19. Int J Infect Dis. 2020; 100: 390-393. doi:10.1016/j.ijid.2020.08.018.
- Importantly, the overall time of this study is very short. Any explanation would convince the readers about the short time period and relevant possible reasons.
Both Astenor Energy and Astenor Forte are presented in the form of an oral solution, which makes their absorption fast, with visible effects already within the first hour of administration. The solution can be administered as is or diluted in water for extra hydration. Also, both formulas can be administered daily, preferably in the first part of the day. It is highly recommended to take it daily, at least 10 days a month. For long-term results, use one of the two forms of ASTENOR, Energy and Forte, over a period of 3 months.
As the leaflet of the products specifies,”the recommended intake period is from 10 days to three months”. For long-term results, it is recommended to use either Astenor Energy or Astenor Forte for a period of 3 months. That is the reason why the period of the study was 3 months and Astenor was prescribed by the doctor, according to the patient's needs.
- The introduction is not sufficient enough to gain the overall background of this study. Adding a bit more relevant literature studies in the introduction will enhance the understanding of the readers.
The authors would like to thank the reviewer for the suggestion to improve the introduction, new references have been inserted in the text of the paper and the introduction was reworded.
Reviewer 2 Report
Comments:
1. From a total cohort of 505 patients, this study collected 146 patients treated with arginine (group 1) or with arginine and malic acid (Group 2) and analyzed the fatigue level using Fatigue Assessment 20 Scale (FAS) through its total (FAS-T), mental (FAS-M) and physical (FAS-P) scores, baseline and at three months of treatment. However, it is not clearly indicated how many of such 146 patients were included in each treatment group. Besides, it is important to highlight the characteristics and severity of the COVID infection in the two groups of the 146 patients included in the clinical trial. On the other hand, it seems that this study would have been enriched by having included a control group treated with placebo.
2. In patients who reported mental or/and physical fatigue symptoms during or after COVID-19 infection, a psychological study would have been carried out prior to inclusion in the clinical trial, for example a depression screening test, and the presence or absence of psychological symptoms prior to COVID infection
3. The definition of physical fatigue (Material and Methods; page 3) includes: “lack of energy for daily activity, decreased muscle strength, muscle pain, sarcopenia”. However, sarcopenia is not a symptom but a sign that must be measured objectively, and as such is not measured in this clinical trial. The presence or absence of muscle pain should be referred to as an independent manifestation of physical fatigue. The results section should indicate how many patients had muscle pain before inclusion in the study, and it should mention whether they improved after 12 weeks of treatment.
4. Apart from the effect of arginine administration alone or combined with malic acid, it would have been highly relevant to have evaluated in these patients the possible existence of other factors (for example, improvement in diet, better and greater night rest, reincorporation to family life and/or work) that may have influenced the perception of fatigue.
Author Response
We are very grateful to the constructive comments from you. We also thank you for the time and effort on reviewing our manuscript. We have carefully addressed point-by-point all the comments and made corrections in our manuscript using tracked changes.
Comments:
- From a total cohort of 505 patients, this study collected 146 patients treated with arginine (group 1) or with arginine and malic acid (Group 2) and analyzed the fatigue level using Fatigue Assessment 20 Scale (FAS) through its total (FAS-T), mental (FAS-M) and physical (FAS-P) scores, baseline and at three months of treatment. However, it is not clearly indicated how many of such 146 patients were included in each treatment group. Besides, it is important to highlight the characteristics and severity of the COVID infection in the two groups of the 146 patients included in the clinical trial. On the other hand, it seems that this study would have been enriched by having included a control group treated with placebo.
We are sorry for the mistake slipped into the Abstract, we verified the number of patients from Results and Table 1 and we found no other mistakes. The characteristics and severity of the COVID infection for the two groups were presented in the Table 1 and before the Table 1.
We did not aim to compare this suplement with a reference product, including placebo. The products (Astenor Energy® and Astenor Forte®) were used in accordance with the terms of the marketing authorisation, and we evaluated if they could be proposed as a remedy to restore physical and mental performance affected by fatigue burden in people with COVID-19 or following COVID-19. This is the reason we compared the fatigue levels before and after the treatment with the two products.
- In patients who reported mental or/and physical fatigue symptoms during or after COVID-19 infection, a psychological study would have been carried out prior to inclusion in the clinical trial, for example a depression screening test, and the presence or absence of psychological symptoms prior to COVID infection.
Thank you very much for this comment. As we stated in the study design, the patients visited the family physician office and the patients’ characteristics were collected. Apart from asking about the BMI, ALT, AST and severity of infection, the comorbidities were also highlighted. As in Table 1, patients with psychological symptoms prior to COVID infection were excluded – the patients were known by their family physicians and there was no need to use a depression screening test. We specified clearer the exclusion criteria for the patients.
- The definition of physical fatigue (Material and Methods; page 3) includes: “lack of energy for daily activity, decreased muscle strength, muscle pain, sarcopenia”. However, sarcopenia is not a symptom but a sign that must be measured objectively, and as such is not measured in this clinical trial. The presence or absence of muscle pain should be referred to as an independent manifestation of physical fatigue. The results section should indicate how many patients had muscle pain before inclusion in the study, and it should mention whether they improved after 12 weeks of treatment.
Thank you very much for this observation. The definition of physical fatigue in page 3 was taken from the original FAS questionnaire [39-44]. We agree with the importance of measuring sarcopenia or muscle pain, and as FAS does not include questions about these symptoms, we excluded them from the definition of physical fatigue.
- Apart from the effect of arginine administration alone or combined with malic acid, it would have been highly relevant to have evaluated in these patients the possible existence of other factors (for example, improvement in diet, better and greater night rest, reincorporation to family life and/or work) that may have influenced the perception of fatigue.
This study had several limitations that are worth noting. We added this paragraph in the discussion section: However, this study did not aim to determine specific nutritional deficiency, but rather to show the general trend of deficiency and the importance of supplementation in cases of COVID-19 infection. An interview before and after 3 months of treatment was also carried out by the family doctors in which they assessed fatigue, BMI, ALT and AST comparing them with the initial values. One of the limitations observed is the fact that the family doctors did not quantify the results from the interviews regarding the improvement of nutrition, and reintegration into family life and work, although these emerged from the fact that the patients after treatment recorded low values of both fatigue and transaminases.
Reviewer 3 Report
This work measured the change in fatigue symptoms with a questionnaire in patients following COVID infection treated with two different arginine based-supplements: group 1 took arginine aspartate, B6 vitamin, biotin, and magnesium, while group 2 took L-arginine and malic acid. After three months of treatment, the fatigue symptoms decreased in both groups. The results are original and interesting, and the manuscript is well-written. However, several issues should be addressed, as described below:
(1) There is confusion or incompatibility between the main objective (to examine the effects of the two arginine-based supplements, line 14) and the study design (a 12-week observational prospective study, line 96). The study was not observational because there was an intervention, i.e., all subjects received treatment, and the outcome variable (fatigue) was assessed before and after the treatment. Moreover, the work measured the change in fatigue levels (or symptoms) but did not measure the effect (or the effect size). More importantly, the study design does not include a control group (without arginine supplementation, either placebo or waiting list) that allows for interpretation of the decrease of fatigue as an effect of the arginine supplementation. All main components of the manuscript should be revised accordingly, from the title to the discussion and conclusions. If the work mentions the effect, a measurement of the effect should be provided, and the interpretation (discussion) and conclusions should be consistent with the actual results provided by the manuscript.
(2) The rationale for comparing two different L-arginine supplementation formulas is not mentioned in the Introduction section.
(3) The objective of the study in the Abstract is not the same as at the end of the Introduction section. Please revise.
(4) The studied groups have many baseline differences, and some of these variables may be confounding factors in comparisons between groups. Why are there so many different variables? Is that the result of some bias in the allocation of patients in each group? Also, How were these baseline differences considered in the interpretation of the comparisons between groups regarding the main variable (i.e., fatigue)? These issues should be addressed in the Methods section as well as in the Discussion section.
(5) What was the purpose of the correlation analysis regarding the main objective of the present work?
Author Response
We are very grateful to the constructive comments from you. We also thank you for the time and effort on reviewing our manuscript. We have carefully addressed point-by-point all the comments and made corrections in our manuscript using tracked changes.
(1) There is confusion or incompatibility between the main objective (to examine the effects of the two arginine-based supplements, line 14) and the study design (a 12-week observational prospective study, line 96). The study was not observational because there was an intervention, i.e., all subjects received treatment, and the outcome variable (fatigue) was assessed before and after the treatment. Moreover, the work measured the change in fatigue levels (or symptoms) but did not measure the effect (or the effect size). More importantly, the study design does not include a control group (without arginine supplementation, either placebo or waiting list) that allows for interpretation of the decrease of fatigue as an effect of the arginine supplementation. All main components of the manuscript should be revised accordingly, from the title to the discussion and conclusions. If the work mentions the effect, a measurement of the effect should be provided, and the interpretation (discussion) and conclusions should be consistent with the actual results provided by the manuscript.
Thank you for your important comment. Our study is an observational, non-interventional study because it respects the definion from REGULATION (EU) No 536/2014 OF THE EUROPEAN PARLIAMENT AND OF THE COUNCIL of 16 April 2014 on clinical trials on medicinal products for human use, and repealing Directive 2001/20/EC: the assessed supplements are authorised and they are used in accordance with the terms of the marketing authorisation (including the leaflet): the fatigue. More, the additional diagnostic (COVID-19 infection) or monitoring procedures do not pose more than minimal additional risk or burden to the safety of the subjects compared to normal clinical practice in any Member State concerned (Romania). We did not prove the efficacy of the supplements for other disease than fatigue and a control group (without arginine supplementation, either placebo or waiting list) was not needed. We demonstrated that the fatigue was due to COVID-19 infection and we measured the levels of fatigue before and after the treatment with the arginine-based supplements, as the title underlights.
(2) The rationale for comparing two different L-arginine supplementation formulas is not mentioned in the Introduction section.
Thank you for your response. We did not compare the outcomes between the two different L-arginine supplementation formulas. We have just underlight the different characteristics of the patients included in Group 1 and Group 2: the patients with high levels of alanine aminotransferase (ALT) and aspartate aminotransferase (AST) were included in Group 1. Also, the patients from Group 2 were significantly more likely to be treated home than those included in Group 1 because of the lower severity of COVID-19 infection.
(3) The objective of the study in the Abstract is not the same as at the end of the Introduction section. Please revise.
Thank you very much for this important comment about the Abstract and the Introduction – we modified it accordingly.
(4) The studied groups have many baseline differences, and some of these variables may be confounding factors in comparisons between groups. Why are there so many different variables? Is that the result of some bias in the allocation of patients in each group? Also, How were these baseline differences considered in the interpretation of the comparisons between groups regarding the main variable (i.e., fatigue)? These issues should be addressed in the Methods section as well as in the Discussion section.
We did not compare the groups, we compared only the fatigue levels, ALT, and AST before and after 3 months of treatment with every supplement. The studied groups have many baseline differences and these differences were the base of including every patient in Group 1 or Group 2. No bias was present in the allocation of patients in each group: as we specified in the manuscript, „As family physicians decided, patients with high levels of alanine aminotransferase (ALT) and aspartate aminotransferase (AST) were included in Group 1 and received for three months, 10 days/month, the treatment with Astenor Energy® (950 mg arginine aspartate, 4 mg vitamin B6, 150 mcg biotin and 83.3 mg magnesium). Patients with normal levels of ALT and AST were included in Group 2 and received Astenor Forte® (413 mg L-arginine and 1500 mg malic acid) for three months, 10 days/month.”.
(5) What was the purpose of the correlation analysis regarding the main objective of the present work?
The correlations proved that the assessed fatigue was dependent on COVID-19 infection severity and no other factors (age, BMI, comorbidities) induced the evaluated fatigue.
Round 2
Reviewer 2 Report
The modifications introduced in the revision of the manuscript are adequate. Many thanks to the authors.
Reviewer 3 Report
No further comments